# Crystallization of Ethylene Plant Hormone Receptor—Screening for Structure

**DOI:** 10.3390/biom14030375

**Published:** 2024-03-20

**Authors:** Buket Rüffer, Yvonne Thielmann, Moritz Lemke, Alexander Minges, Georg Groth

**Affiliations:** Institute of Biochemical Plant Physiology, Faculty of Mathematics and Natural Sciences, Heinrich Heine University Düsseldorf, 40204 Düsseldorf, Germanyyvonne.thielmann@hhu.de (Y.T.); alexander.minges@hhu.de (A.M.)

**Keywords:** integral membrane proteins, plant hormone receptor, ethylene receptor histidine kinase, high-affinity copper binding, Cu(I) metal cofactor, transmembrane domain (TMD), biological function, metal-dependent ligand binding, LCP crystallization, protein structure

## Abstract

The plant hormone ethylene is a key regulator of plant growth, development, and stress adaptation. Many ethylene-related responses, such as abscission, seed germination, or ripening, are of great importance to global agriculture. Ethylene perception and response are mediated by a family of integral membrane receptors (ETRs), which form dimers and higher-order oligomers in their functional state as determined by the binding of Cu(I), a cofactor to their transmembrane helices in the ER-Golgi endomembrane system. The molecular structure and signaling mechanism of the membrane-integral sensor domain are still unknown. In this article, we report on the crystallization of transmembrane (TM) and membrane-adjacent domains of plant ethylene receptors by Lipidic Cubic Phase (LCP) technology using vapor diffusion *in meso* crystallization. The TM domain of ethylene receptors ETR1 and ETR2, which is expressed in *E. coli* in high quantities and purity, was successfully crystallized using the LCP approach with different lipids, lipid mixtures, and additives. From our extensive screening of 9216 conditions, crystals were obtained from identical crystallization conditions for ETR1 (aa 1-316) and ETR2 (aa 1-186), diffracting at a medium–high resolution of 2–4 Å. However, data quality was poor and not sufficient for data processing or further structure determination due to rotational blur and high mosaicity. Metal ion loading and inhibitory peptides were explored to improve crystallization. The addition of Zn(II) increased the number of well-formed crystals, while the addition of ripening inhibitory peptide NIP improved crystal morphology. However, despite these improvements, further optimization of crystallization conditions is needed to obtain well-diffracting, highly-ordered crystals for high-resolution structural determination. Overcoming these challenges will represent a major breakthrough in structurally determining plant ethylene receptors and promote an understanding of the molecular mechanisms of ethylene signaling.

## 1. Introduction

Integral membrane proteins play crucial roles in cellular signaling and information transfer across biological membranes. These structures recognize signaling molecules ranging from small gaseous molecules to large compounds. Despite their great importance for intra- and intercellular communication and adaptation to environmental stimuli and stresses, our knowledge of the structure and biophysical properties of many receptor proteins is still limited [1,2,3].

The gaseous hormone ethylene regulates a wide range of essential functions in plant growth and development and is a well-known mediator of stress responses [1,4,5,6]. Signal perception and response to the plant hormone ethylene have been extensively studied in the model plant *Arabidopsis thaliana*. These studies have identified several mutants and the related molecular components involved in ethylene signaling [1,7,8,9]. Phenotypic, molecular, and biochemical analyses suggest that ethylene perception and signaling are mediated by a family of integral membrane receptors consisting of Ethylene response 1 (ETR1) and its four isoforms ERS1 (Ethylene response sensor 1), ETR2 (Ethylene response 2), ERS2 (Ethylene response sensor 2), and EIN4 (Ethylene insensitive 4). Receptors ETR1 and ERS1 belong to subfamily I, while the other three isoforms form subfamily II [10,11,12]. All members of the ethylene receptor family have a similar architecture, including a transmembrane domain (TMD) that contains three helices in subfamily I (TMH1-3) and four helices in subfamily II (TMH1-4). The TMD is critical for receptor dimerization and high-affinity binding of Cu(I) cofactors, which are essential for ethylene binding and signal transfer. The TMD is followed by cGMP-specific phosphodiesterase, adenylyl cyclase, and FhlA (GAF) domains responsible for receptor dimerization and the formation of high oligomeric clusters [13,14,15]. Other domains include the histidine kinase (HK) domain and the receiver domain (RD). The structure of individual cytosolic subdomains of the ETR family, which are similar to bacterial two-component histidine kinases [16], has been identified by X-ray crystallography and small-angle X-ray scattering (SAXS) [17,18,19].

Crystallization and structural studies of full-length receptors or isolated TMD have not succeeded to date. Compared to soluble proteins, crystallization of membrane proteins is notoriously difficult and typically involves extracting the target from its native lipid membranes. In this process, commonly used detergents cover most of the membrane part (hydrophobic region), leaving only a small surface area (loops and the hydrophilic region) of the target protein for crystal contacts. Moreover, these crystals are often fragile and have low-resolution diffraction and crystallization defects. In addition, optimizing crystal packing with detergents is very time-consuming [20]. Membrane protein crystallization with LCP, where membrane proteins such as ETRs are embedded in a membrane-mimicking environment, is more suitable for high throughput screening. This method allows hydrophobic lipid-guided interactions and hydrophilic protein–protein interactions, leading to tight crystal packing. The first crystallization approaches with LCP were called bicontinuous cubic phases by Landau and Rosenbusch [21]. They intended to build a structured and flexible system where membrane proteins are incorporated into membranes using monoolein (MO) or monopalmitolein (MP). Crystals grow in three-dimensional space. In this method, called *in meso* or LCP crystallization [22], membrane proteins are mixed in detergent micelles with a lipid or lipid mixture in a specific ratio to form a homogenous mixture of reconstituted membrane proteins. Further development of *in meso* crystallization is sponge phase crystallization [23]. The solvent diameter of the solvent channels is enlarged and the high order of the cubic phase is distorted, making this method more suitable for membrane proteins with a large extracellular domain [23,24].

Another approach to enlarging the solvent channels is to use Cherezov and Caffrey’s widely applied method, where the channel pores of the membrane protein of interest are occupied by different lipids or lipid mixtures, e.g., MO, MP, or MO with the addition of cholesterol or cardiolipin (Figure 1) [25]. An LCP lattice formed by MO alone has a lattice parameter of 106 Å. The lattice can be enlarged with cholesterol and cardiolipin to about 170 Å and 190 Å (Figure 1). Lattice expansion was further developed by Zabara et al. with an MP/1,2-distearoyl-sn-glycerol-3-phosphoglycerol (DSPG) mixture, which retains more water in the LCP system and forms a lattice with a maximum lattice parameter of 525 Å (Figure 1) [26].

In recent years, LCP has enabled the crystallization of a wide range of membrane proteins from enzymes [27] to transporters, channels, and receptors to structural proteins and complexes [28,29,30,31,32,33,34,35]. Its success prompted us to explore this technology for ETR1 and to foster LCP crystallization of members from the two subfamilies using different additives, screening kits, and lipid mixtures. Focusing on the structure of the TM and GAF domains, we used truncations ETR1^1-157^, ETR1^1-316^, ETR1^1-407^, and ETR2^1-186^ for *in meso* crystallization. This article reports our progress on the structural analysis of the TM and GAF domains.

## 2. Materials and Methods

### 2.1. Materials

Chemicals and reagents were purchased from AppliChem (Darmstadt, Germany), Glycon (Luckenwalde, Germany), VWR International (Geldenaaksbaan, Belgium), BD (Le Pont de Claix, France), Carl Roth (Karlsruhe, Germany), BIOZOL (Eching, Germany), LIPOID (Dortmund, Germany), Jena Bioscience (Jena, Germany), Cytiva (Marlborough, MA, USA), Merck Millipore (Burlington, MA, USA), Miltenyi Biotec B.V. & Co. KG (Bergisch-Gladbach, Germany), and Molecular Dimensions (Rotherham, UK) at analytical grade. The pET16b-plasmid was purchased from MERCK/Novagen (Darmstadt, Germany), and the pGEX4T-1 plasmid was purchased from GE Healthcare (Munich, Germany). Oligonucleotides were synthesized by Sigma–Aldrich/MERCK (Steinheim, Germany). Two peptides, nuclear localization signal octapeptide 1 (NOP-1) and NLS icosapeptide 1 (NIP-1), were synthesized by Genscript (Piscataway, NJ, USA). For crystallization trial setups, ProCrysMeso (Zinsser Analytics, Eschborn, Germany) was used. A SterREO Discovery V.12 binocular (Zeiss, Oberkochen, Germany) equipped with a UV detector XtalLight100 (Xtal Concepts, Hamburg, Germany) was used for crystal detection.

### 2.2. Methods

#### 2.2.1. Cloning, Heterologous Expression, and Purification

The pGEX4T-1 *ETR2^1-186^ mT2* 10x His plasmid was derived from the pGEX4T-1 *ETR2* expression vector, as previously described [13]. The construct was truncated to remove amino acids (aa) 187-773. In addition, the fluorophore mCerulean was fused to the receptor’s 10x His-tag and further modified by mutagenesis at positions T65, A145, and I146 to obtain *mTurquoise2* (mT2, see Figure 2), increasing the fluorescent reporter’s brightness and photostability [36]. The primers used in this process are listed in the SI (Appendix A). The pGEX4T-1 *ETR1^1-157^ mT2* 10x His plasmid was derived from the pGEX4T-1 *ETR2^1-186^ mT2* 10x His expression vector, as previously described (Figure 2) [13]. The coding sequence for *ETR1^1-157^* was obtained from the pETEV16b *AtETR1* expression vector by removing nucleotides encoding the soluble part of the receptor (aa 158-738, Figure 2). The pETEV16b *ETR1^1-316^* plasmid was derived from the pETEV16b *AtETR1* expression vector (Figure 2) by deleting the coding sequence for aa 317-738. The pETEV16b *ETR1^1-407^* plasmid was used as previously described (Figure 2) [37]. The sequences of all plasmids used in this study were confirmed by sequencing with T7 and T7 terminator primers. For heterologous expression, pETEV16b *AtETR1^1-316^*, pETEV16b *AtETR1^1-407^*, pGEX-4T-1 ETR1^1-157^ mT2, or pGEX-4T-1 ETR2^1-186^ mT2 were transformed into *E. coli* C41(DE3)∆(*ompF*-*acrAB*) [38] and grown on 2YT agar plates containing 100 µg/mL of ampicillin at 37 °C overnight. Pre- and main cultures of pGEX-4T-1 *ETR2^1-186^ mT2*, pGEX-4T-1 *ETR1^1-157^ mT2*, *ETR1^1-407^*, and pETEV16b *AtETR1^1-316^* were prepared as previously described for the expression and purification of ETR1^1-157^ [39]. pETEV16b *AtETR1^1-316^* was expressed at 30 °C and 16 °C [39]. Immobilized metal affinity chromatography (IMAC) of AtETR1^1-316^ and ETR2^1-186^ mT2 was conducted as described in [40]. HEPES buffer (50 mM HEPES, pH 8.0, 200 mM NaCl, 0.015 (*w*/*v*) % FosCholine 14, 0.002 (*w*/*v*) % PMSF) was used for IMAC purification of ETR2^1-186^ mT2. After buffer exchange, the purified protein was concentrated to a volume ≤ 500 µL with a concentration ≥10 mg/mL. Aliquots of 60–80 µL were shock frozen with liquid nitrogen and stored at −80 °C or used directly for crystallization trials. The purity and homogeneity of all samples were analyzed and characterized by SDS-PAGE and immunodetection.

#### 2.2.2. LCP Crystallization

Crystallization and structural studies of detergent-solubilized ethylene receptors and isolated TMDs have not been successful. Therefore, we pursued LCP crystallization as an alternative strategy to provide a more biologically native lipid environment for the ETR1 TMD. Unlike detergents, which disrupt ETR1 structure, particularly in the membrane, LCP is thought to preserve TMD structure, stability, and functionality. Compared to detergent-based crystals, LCP crystals have enhanced crystal order and diffraction quality due to their reduced solvent content. As a result, high-resolution experimental structural information can be obtained from isolated receptor proteins. LCP crystallization or *in meso* crystallization is a method where proteins are crystallized in a membrane-mimicking environment. The general setting for LCP crystallization is a protein/lipid mixture at a defined ratio mixed in a coupled syringe system to form LCP [35]. Typically, LCP crystallization experiments are conducted in a glass sandwich setup consisting of two glass plates and a spacer. In such a setup, the LCP mixture is placed onto the lower glass plate and overlaid with a precipitant solution (Figure 3A). The upper glass plate is then placed on top of the spacer, sandwiching the LCP batch experiment between the plates (Figure 3A). To harvest the growing crystals from the LCP, the upper glass must be cut before the crystals can be removed with a loop (see Figure 3A). To facilitate handling, we used regular sitting drop plates in our setup (MPI tray [43], Figure 3B) as previously established in [44,45]. However, it should be noted that any commercially available round-bottomed sitting drop plate can be used for this protocol. The LCP (100 nL) was dispensed into the protein well and overlaid with 1.5 µL of precipitant solution (Figure 3B). The reservoir well was then filled with 35 µL of precipitant solution (Figure 3B). In this way, LCP crystallization is no longer a static batch experiment but features a diffusion-driven component. Another important benefit of the microplate setup used is that it greatly facilitates crystal harvesting. No glass cover is cut when the plate is sealed with adhesive UV-compatible foil. Previous studies showed that with such a setup, crystals can be harvested from the plates even after nine months and diffracted up to 2.0 Å [44].

#### 2.2.3. LCP—Lipid Mixture Preparation

Monoacylglycerols monoolein (MO, MAG 9.9) and monopalmitolein (MP, MAG 9.7) were heated up to 42 °C. Then, 0.12 g of the liquefied lipid was pipetted into glass vials. Thereafter, 0.013 g of lipid additives 1,2-dipalmitoyl-3-succinylglycerol (DSPG), cholesterol, 1,2-dioleoyl-sn-glycerol-3-phosphocholine (DOPC), or 57 mg cardiolipin in chloroform were added alongside 300 μL of chloroform. These lipid–chloroform mixtures were shaken frequently for 30 min at 42 °C until the lipids formed a homogeneous solution. Chloroform was removed from the glass vials under a fume hood with a stream of air. To remove the remaining solvents, the lipid mixtures were vacuumed overnight. All lipid samples were stored at −20 °C or used directly for sample preparation.

#### 2.2.4. LCP—Sample Preparation

Normally, LCP crystallization was performed with a ratio of 60% lipid and 40% protein solution. Diverging from standard conditions, ratios of 70:30% and 50:50% (protein:lipid) were used for MP/DSPG and MP/cholesterol or MP/DOPC, respectively. For ETR2^1-186^ mT2 and ETR1^1-157^ mT2, the LCP was formed using the lipids MO, MO/DSPG, MO/cholesterol, MP/DOPC, and MP/DSPG. For this purpose, ETR1^1-316^ and ETR1^1-407^ were mixed with MO/DSPG, MO/cardiolipin, MO/cholesterol, and MP/cholesterol to obtain lipid mixtures for an LCP with enlarged lattice parameters (Figure 1) [34]. Lipid-filled glass vials were heated to 42°C. Molten lipid was filled into one Hamilton syringe of the coupled syringe system used for LCP mixtures (Innovative Labor System GmbH, Stützerbach, Germany). Thawed protein was filtered through a 0.2 µm filter (Merck Millipore, Ultrafree-MC-GV-Centrifugal-Filter-Units, MA, USA) to remove aggregates. Lipid mixtures and protein solutions were mixed to obtain homogeneity. These LCP mixtures were pipetted onto MPI trays using the ProCrysMeso robot with humidification. The bolus contained 100 nL of LCP and was overlaid with 1.5 µL of crystallization solution (Figure 3B). The reservoir well was filled with 35 µL of crystallization solution (Figure 3B) [43]. Plates were covered with ClearVue Sheets (Molecular Dimensions, Rotherham, UK) and stored at 22 °C. In our study, crystallization screens (MemMeso HT-96, MemGold1 HT-96 Eco Screen, MemGold2 HT-96 Eco Screen, MemTrans Eco, MemChannel Eco, XP screen, BCS screen Eco, The Cubic Phase I Suite, The Cubic Phase II Suite, MIDAS, Structure screen 1 CF and 2, MemSys, MemStart and MemPlus Eco, MemStart), customized screens at pH 7.0 and 8.5, and a citrate screening kit [46] were used. In addition to various screen solutions, 2 mM of inhibitory peptides NIP-1 [37] or NOP-1 [47,48], 1 mM ZnCl_2_, 100 µM of ammonium molybdate or 5 mM of EGTA with 50 mM ß-mercaptoethanol were added to the crystallization setup. ETR2^1-186^ mT2 was saturated with Cu-BCA prior to LCP preparation to fully load the receptor with its monovalent copper cofactor, as described by Schott-Verdugo et al. [39].

## 3. Results and Discussion

In recent years, LCP has enabled the crystallization of a wide range of membrane proteins [44,45,49,50]. This success prompted us to explore this technology, particularly in the transmembrane and membrane-adjacent regions of ethylene receptors ETR1 and ETR2, which harbor the plant hormone binding site and the monovalent copper cofactor essential for biological function. To this end, we cloned and expressed isolated subdomains and subdomain fusions of *ETR2* in a bacterial host. Of these, *ETR2^1-186^ mT2*, *ETR1^1-157^ mT2*, *ETR1^1-407^*, and *ETR1^1-316^* were successfully expressed in bacterial cells after chemical induction with isopropyl-β-D-1-thiogalactopyranoside (IPTG). Purification of all four protein constructs after overnight expression resulted in large amounts (0.5–2 mL) of pure (see protein gels in Figure 4) and homogeneous material (ETR1^1-407^ 16 mg/mL; ETR1^1-316^ 19 mg/mL; ETR1^1-157^ mT2 16 mg/mL; ETR2^1-186^ mT2 11 mg/mL). Analysis of these preparations by SDS-PAGE and Western blotting (Figure 4) revealed only minor impurities, indicating that samples contained related monomers and SDS-stable dimers. For ETR1^1-157^ mT2, a minor degradation product of ~20 kDa was observed in addition to the correct TMD monomer. For LCP crystallization of the purified recombinant ETR1 and ETR2 protein constructs, MO and MP were applied as standard lipids. Considering the different molecular mass and membrane-adjacent extent of the four constructs, further lipids were added to the standard setup to expand the lattice parameters of the LCP (Figure 1) [25]. The addition of anionic phospholipid DSPG to the LCP mixture increases solvent channels in the LCP to 26.8 nm in diameter through electrostatic and steric remodeling [51]. By contrast, when MO is used on its own, solvent channels are limited to ~12 nm through electrostatic swelling in a sponge phase [25]. To mimic natural membrane cholesterol composition, CHS, cardiolipin, or DOPC have also been used as additives to MP or MO [25,52,53]. In particular, DOPC, expected to be a promising additive to this lipid together with other phosphatidylcholines, is highly abundant in the ER endomembrane system where ETRs reside in the plant [54,55,56]. In addition to their effect on lattice parameters, doped lipids can form cubic phases with different geometries and water channel sizes, respectively (Figure 1) [25]. Finally, LCP formation also depends on the length and branching of the lipids as well as on the water content of the lipid mixture used. While cholesterol and DOPC only slightly extend the lattice parameters, cardiolipin and DSPG form extended water channels [22] suitable for membrane proteins with large extracellular domains such as the *Gloeobacter violaceus* ligand-gated ion channel protein [26] or, in our case, ETRs (Figure 2).

Quantitative evaluation of our screening trials revealed a higher number of crystal hits for MO/cholesterol MP/DOPC than corresponding setup trials for MO alone or other lipid mixtures (Figure 5 and Appendix A). In addition to the difference in number, crystals from MO-lipid mixtures also appeared more rapidly (1–7 days) than in corresponding screenings with pure MO or MP-lipid mixtures (2–12 weeks).

Depending on the lipid crystals obtained in our LCP screening, trials showed significant variations in size (35–200 µm) and morphology from amorphous crystals to cubes, needles, plates, and rods (Figure 5). Most of these crystals showed no diffraction on synchrotron beamlines, except for crystals from ETR1^1-316^ and ETR2^1-186^ mT2, which diffracted up to 4 Å and 2 Å, respectively. However, as shown in Figure 6, the overall data quality for these crystals was poor. The related diffraction pattern was rotationally blurred and showed high mosaicity, indicating disorders and poorly defined lattice packing. Peak integration and processing of the data collected for these crystals failed. Attempts to further optimize the initial crystallization conditions from the screening (0.1M CaCl_2_, 0.1 M Tris pH 8.5; 28% (*v*/*v*) PEG300) by varying the salt or PEG concentrations have not succeeded. As a result of high salt, crystal processing and analysis were further complicated by salt crystals forming alongside the protein crystals, which interfered with harvesting protein crystals from the LCP and caused disturbing background noise in the diffraction images (Figure 6). By contrast, only non-diffracting protein crystals were obtained with low salt.

In the past, several crystallization studies revealed the significant role of protein stabilization in obtaining diffraction-quality crystals. To this end, target proteins are complexed with substrates, nucleic acids, cofactors, or small molecules. Successful examples include the bacterial two-component HK, a protein family closely related to HK-related ETRs. Similar to ETRs, many of these proteins contain a sensor domain linked to the cytoplasmic kinase module via a transmembrane structure. While bacterial HKs’ sensor domains are typically located in the periplasm, ETRs’ sensor domains are fully integrated into the transmembrane structure. The sensor, transmembrane, and membrane adjacent HAMP domains of the nitrate/nitrite sensor kinase NarQ from *E. coli* have been solved by *in meso* crystallization and single-wavelength anomalous diffraction approaches in the ligand-bound form [57,58]. Structural alignment with the ligand-free apo structure of NarQ revealed that nitrate binding in the sensor domain triggers substantial rearrangements in the transmembrane structure, which are thought to reflect molecular events in HK signaling. Further examples of stabilized and structurally resolved bacterial HKs relate to the sensor domains of receptor kinases CusS and NarX from *E. coli* and CitA from *Klebsiella pneumoniae* [59,60,61], all of which require their stimulus (nitrate for NarX, Cu(I) for CusS and citrate for CitA) as cofactors for receptor dimerization. Dimerization is thought to induce conformational changes in the transmembrane structure, which are transferred to the kinase module.

A well-known cofactor of ETRs that is essential for their biological function is monovalent copper, which is bound to their transmembrane structure. Current data suggest that the metal cofactor is not required for receptor dimerization, but rather plays an essential role in providing a high-affinity binding site for the plant hormone ligand [13,37,62]. Studies of purified ETR1 reconstituted into unilamellar liposomes by EPR spectroscopy indicate that copper loading on the receptor does not cause major conformational changes in the transmembrane structure. However, given the spatial resolution of 1–2 nm for this technique [63], ligand-induced rearrangements of a few Angstroms, as observed for bacterial HKs [59,60,61], cannot be completely excluded at this stage. Therefore, to arrest the receptor in a defined and homogenous state, the copper loading state of the purified recombinant ETR must be fixed. Previous studies in our lab showed that about 20% of recombinant ETR1 produced in *E. coli* is preloaded with Cu(I) from the bacterial host. Therefore, to obtain a uniform, fully loaded preparation for LCP crystallization, recombinant ETRs were preloaded with Cu-BCA, as described previously [39,64]. Alternatively, Cu(I) was completely removed from the ETRs according to protocols described for the copper-exported P-type ATPase CopA [65]. In addition to their natural metal cofactor, purified recombinant proteins can bind to other metal ions that can maintain or even stabilize their structure. In this sense, Cu(I) has been replaced by Zn(II) in human and cyanobacterial copper chaperones for structural studies [66,67,68]. Thus, in addition to fixing the copper loading state of our purified ETRs, we also used zinc as a replacement in our LCP crystallization trials. In summary, our controlled metal-loading experiments revealed that many well-formed protein crystals are formed upon Zn(II) and Cu(I) addition by Cu-BCA. Conversely, the removal of the metal cofactor with TTM/EGTA/β-mercaptoethanol did not affect crystallization (Appendix A).

Previous studies from our laboratory [37,47,48,69] showed that the small synthetic peptide NOP-1 (LKRYKRRL), corresponding to an interaction sequence (NLS) in a downstream ETR binding partner, tightly binds to the ETR1 GAF domain and probably prevents conformational changes, leading to increased structural stability of the receptor dimer [37]. When applied to the plant, the peptide showed visible effects on plant ethylene responses. NOP-1 successfully delayed ripening and senescence in tomatoes [48,70], broccoli [71], and apples [72] by six to eight days. Binding studies by MST on purified ETR receptors from tomato, apple, and *Arabidopsis* demonstrated the peptide’s high affinity for binding to receptors in the range of 80–100 nM [37,48,70,72]. Extending the sequence of the NOP-1 inhibitory peptide with additional residues (12 aa) adjacent to the NLS binding motif in the EIN2 downstream interaction partner (NIP peptide) further improved binding affinity [37]. Both inhibitory peptides were used for co-crystallization of our ETR constructs in the LCP screening trials to stabilize their transmembrane and membrane-adjacent domains. While the addition of NIP resulted in many well-formed rods and needles, NOP-1 addition had less effect on crystallization. However, compared to the previous metalation of the receptor by Cu(I) or Zn(II), the impact of both peptide ligands was less pronounced (Appendix A).

A total of 9216 crystallization trials were tested with four ETR constructs, lipids (MO, MP, cholesterol, cardiolipin, DSPG, DOPC), ligands (Cu-BCA, ZnCl_2_, ammonium molybdate/EGTA/β-mercaptoethanol, NIP, NOP), and 18 different crystallization kits (for details see Material and Methods). For ETR2^1-186^ mT2, more than 3900 crystallization conditions were screened with 17 crystallization kits, additives Cu-BCA and ZnCl_2_, and lipid combinations MP/DSPG, MO/DSPG, MO, and MO/DOPC. For ETR1^1-316^, lipid combinations MO/DSPG, MO/cholesterol, MO/cardiolipin, and MP/cholesterol, additives NIP, NOP, TTM/EGTA/β-mercaptoethanol, and 11 crystallization kits were used. Over 3700 crystallization conditions were screened. For ETR1^1-157^ mT2, lipids (MO and MO/cholesterol) were used with six crystallization kits, resulting in 768 different crystallization conditions. Similarly, 768 crystallization conditions were tested for ETR1^1-407^, which was applied using the additive NIP, seven crystallization kits, and lipid combinations MO/DSPG and MO/Cardiolipin. A total of 566 ETR crystals were screened at high-performance synchrotron beamlines (Appendix A). The formation of crystals was observed under different conditions; however, in most cases, they exhibited no or poor diffraction, possibly due to the intrinsic flexibility of ETR domains and resulting disorders in the crystal lattice. Attempts to restrict ETR flexibility by adding metal cofactors or inhibitory ligands have not succeeded.

## 4. Conclusions

Crystallization in lipid mesophases is a useful approach to studying and resolving membrane protein structures in a lipid-like environment. Plant ethylene receptors are a plausible target for this technique as the structure of their transmembrane sensor domain has not been resolved by other means yet. The TM domain of ethylene receptors ETR1 and ETR2, which is expressed in *E. coli* in high quantities and purity, was successfully crystallized using the LCP approach with different lipids, lipid mixtures, and additives. Of the many crystals obtained in our extensive screening, only two conditions provided crystals of ETR1^1-316^ and ETR2 ^1-186^ mT2 with clear but blurred diffraction up to 4 Å and 2 Å, respectively. Although we tested known metal cofactors and inhibitory peptides of the receptors as additives under these conditions, we have not obtained well-diffracting highly-ordered crystals of these ETR structures. We assume that the high intrinsic flexibility of the TMD, which is supported by previous EPR studies, hampers high-resolution diffraction over a broad rotation angle. Therefore, TMD flexibility should be restrained for further crystallization attempts. Possible strategies include intra- and intermolecular cross-linking, termini-restraining, or the use of orthologous ETR1 receptors from different species [73,74,75]. Increased stability and rigidity of the TMD, as well as the rigidity of the receptor’s membrane-adjacent domains, should improve structural determination by NMR, cryo-EM, or X-ray crystallography.

## Figures and Tables

**Figure 1 biomolecules-14-00375-f001:**
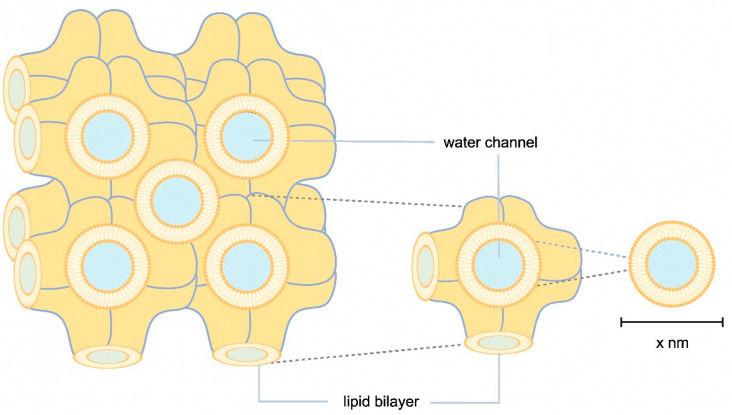
Schematic illustration of LCP. In orange, lipid bilayers are illustrated in highly convoluted membranes, where membrane proteins are embedded through hydrophobic interactions. Water channels, shown in blue, transport the precipitant. The size of the water channels can vary depending on the lipids, stoichiometry, additives, and precipitant used.

**Figure 2 biomolecules-14-00375-f002:**
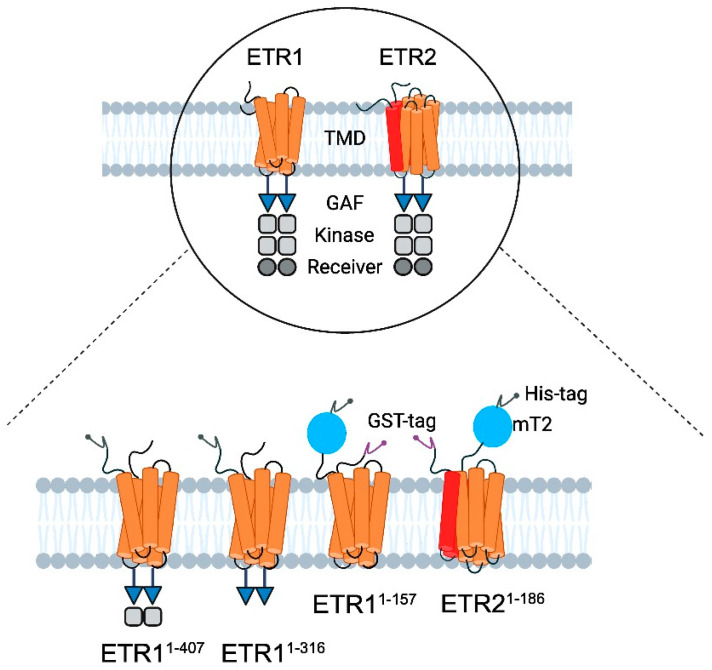
Schematic illustration of the modular structure of ethylene receptors and constructs used for LCP crystallization. (**Top**) ETR1 and ETR2 are two of five ethylene receptor isoforms located in the ER membrane of *Arabidopsis thaliana* [41]. They share a similar modular structure starting at their N-terminus with three transmembrane helices (orange) per monomer in ETR1. By contrast, ETR2 has a putative fourth helix (red) in the TMD monomer [1,7]. In both isoforms, the TMD is followed by the GAF domain (blue triangle), kinase domain (grey squares), and receiver domain (dark gray circle) [1]. The kinase domain of ETR2 is degenerated, and its serine–threonine kinase activity contrasts with ETR1’s histidine kinase activity [1,42]. The receptors only function as dimers [1,14]. (**Bottom**) For purification and crystallization, ETR1^1-316^, ETR1^1-407^, ETR1^1-157^ mT2, and ETR2^1-186^ mT2 were used. All constructs contain a 10x His-tag (dark gray, N-terminal—ETR1^1-316^, ETR1^1-407^; C-terminal—ETR1^1-157^ mT2 and ETR2^1-186^ mT2). ETR1^1-157^ mT2, and ETR2^1-186^ mT2 are additionally flanked at the N-terminus by a GFP—derivative mTurquoise2 (blue) and GST tag (purple, N-terminal).

**Figure 3 biomolecules-14-00375-f003:**
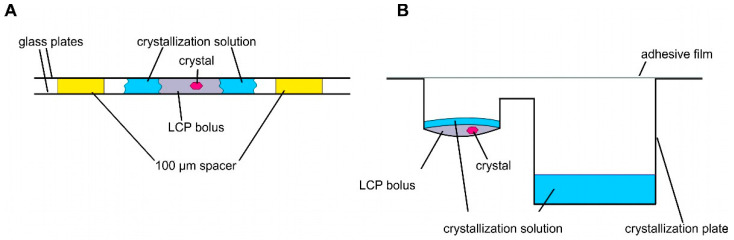
Schemes of different LCP crystallization approaches. (**A**) Sandwich LCP approach: the LCP bolus is covered with a crystallization solution and sandwiched between two glass plates. Spacers of 100 µm keep the glass plates apart. (**B**) Sitting drop LCP crystallization in a round-bottomed sitting drop crystallization plate. An LCP bolus is pipetted into the round cavity covered with crystallization solution. The crystallization solution is stored in the reservoir (rectangular cavity).

**Figure 4 biomolecules-14-00375-f004:**
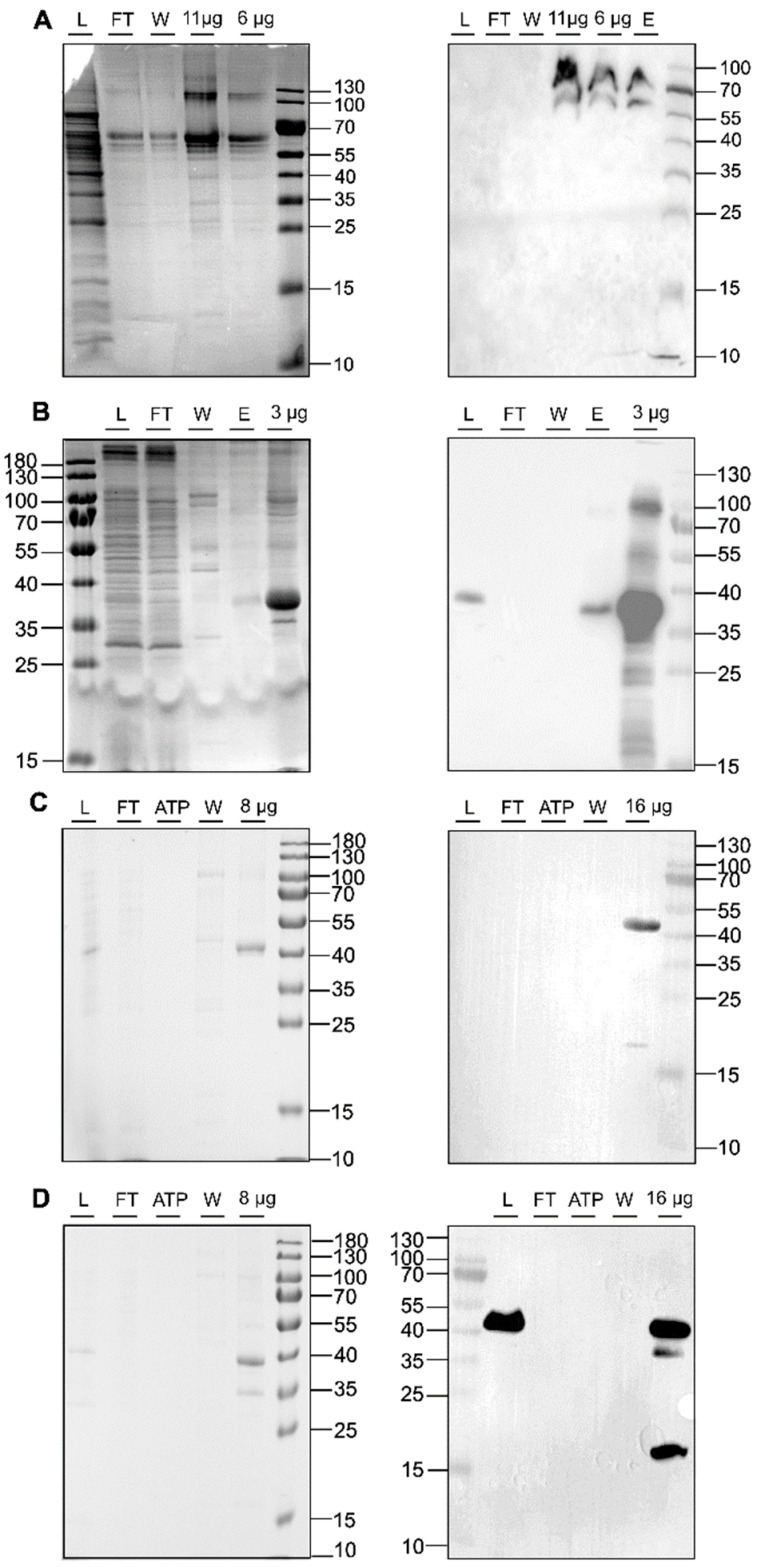
SDS-PAGE protein gels (**left panel**) and Western blots (**right panel**) of IMAC purifications for ETR2^1-186^mT2 (**A**), ETR1^1-316^ (**B**), ETR1^1-407^ (**C**), and ETR1^1-157^mT2 (**D**). The blots and gels were marked with sample load (L), flow-through (FT), ATP wash step (ATP), 50 mM imidazole wash step (W), 250 mM elution step (E), or x µg of the concentrated sample after imidazole removal. For immunoblotting, anti-GFP-HRP (Santa Cruz Biotechnology, Danvers, MA, USA) was used (**A**). Western blot signals in (**B**–**D**) were detected by HRP-conjugated anti-His antibody (Miltenyi Biotec B.V. & Co. KG, Bergisch-Gladbach, Germany). Colloidal Coomassie staining was used for SDS_PAGE protein gels shown on the left panel.

**Figure 5 biomolecules-14-00375-f005:**
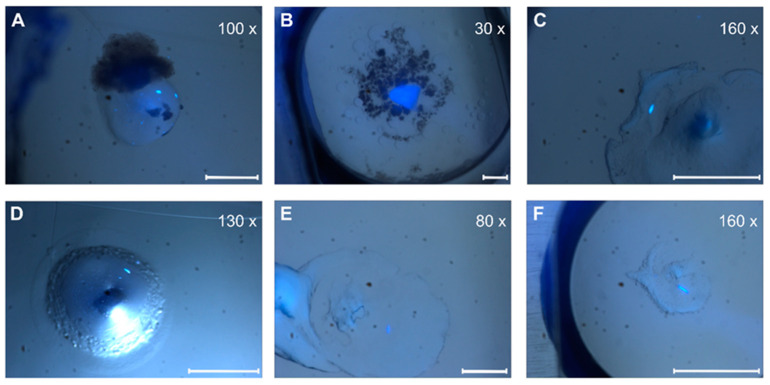
Crystal pictures under UV light. (**A**,**D**) ETR1^1-316^; (**B**,**E**) ETR1^1-407^; (**C**,**F**) ETR1^1-157^ mT2. Depending on the magnification factor, the white scale bars represent a 100 µm length.

**Figure 6 biomolecules-14-00375-f006:**
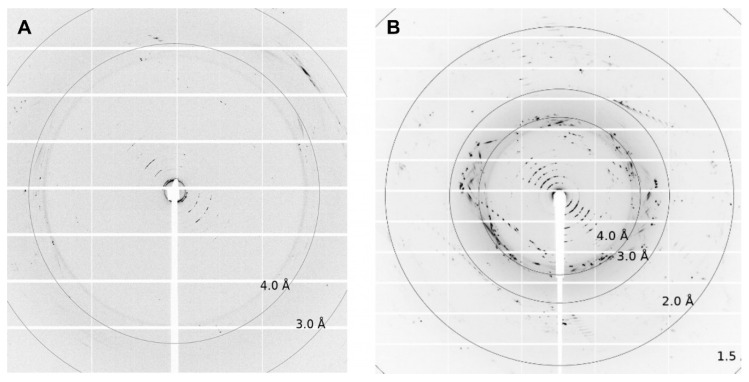
Diffraction pictures of (**A**) ETR1^1-316^ and (**B**) ETR2^1-186^ mT2. For ETR1^1-316^, single spots up to 4 Å can be detected. For ETR2^1-186^ mT2, diffraction spots reach up to 2Å. The crystallization conditions used for both crystals were 0.1 M CaCl_2_, 0.1 M Tris pH 8.5; 28% *v*/*v* PEG300.

## Data Availability

Data is available from the authors upon reasonable request.

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
