# Peer review of "Crystallization of Ethylene Plant Hormone Receptor—Screening for Structure"

_biomolecules, 2024, doi:10.3390/biom14030375_

Round 1

Reviewer 1 Report

Comments and Suggestions for Authors

Dear authors

Thank you so much for your manuscript and your efforts.

Key words: should be summarized to be 4 to 5 words.

Abstract

·   While the abstract describes the crystallization process and challenges faced, a brief mention of any initial findings or insights gained from the crystals, even if preliminary, could add depth to the abstract.

·   The mention of crystals for ETR1 (aa 1-316) and ETR2 (aa 1-186) is made, but specifying if these crystals were obtained under different conditions or if they share similarities in their formation would enhance clarity.

·   The mention of poor data quality and insufficiency for further structure determination could be elaborated by briefly discussing specific issues encountered.

·   It would be beneficial to conclude the abstract with a statement about the significance of overcoming these challenges and a brief mention of potential future steps in the research.

Introduction

·  The first paragraph should be cited.

·  I noticed that there are many paragraphs without references.

·  The inclusion of figures or diagrams illustrating the ethylene receptor architecture, LCP crystallization process, or relevant molecular structures could improve the visual appeal and clarity of the abstract.

·   Please give a more seamless transition to the specific focus of the study, which appears in the latter part of the text.

·   Including schematic figures or diagrams illustrating the structural components of ethylene receptors and the in meso crystallization techniques would enhance visual understanding.

·   Explaining acronyms such as TMD and GAF upon first mention to ensure accessibility for readers who may not be familiar with the terminology.

·   Increase the connection between the introduction and the methods section to explicitly outline how the challenges mentioned in the introduction are addressed using the in meso crystallization technique.

Material and methods

·  Providing a brief discussion on how the proposed strategies for restraining TMD flexibility align with or differ from existing approaches in the literature.

·  Demonstrating any specific experiments or methodologies that will be employed in the proposed strategies for restraining TMD flexibility.

·  If possible, to give more good photos in figure 4 especially (E, F and G).

Results and discussion

·  Explain whether further optimization strategies are planned or if there are alternative approaches being considered.

·  If applicable, discuss any insights gained from the unsuccessful trials and their implications for future experiments.

·  Give potential reasons for the observed high mosaicity and rotational blurring, providing insights into the underlying structural disorder.

·  Explain the potential future directions and discuss the insights gained from the unsuccessful trials will further enrich the narrative.

·  It should add a brief discussion on potential future directions or implications of the findings.

·  Ensure clarity in the presentation of results, especially in Figure S1, to facilitate reader understanding.

Conclusion section:

·   Provides a brief discussion on how the proposed strategies for restraining TMD flexibility align with or differ from existing approaches in the literature.

·   Demonstrate any specific experiments or methodologies that will be employed in the proposed strategies for restraining TMD flexibility.

·   In conclusion, it should provide a detailed comparison with other methods or studies in the field, which could contextualize the significance of the findings and the novelty of the approach more effectively.

References section

·        There are some old references that should be updated (if possible 5 recent years)

Reviewer 2 Report

Comments and Suggestions for Authors

Opinion about the Manuscript ID: biomolecules-2888780

Title: «Crystallization of Ethylene Plant Hormone Receptor – Screening for the Structure».

Authors: Buket Rüffer, Yvonne Thielmann, Moritz Lemke, Alexander Minges, Georg Groth*.

The ethylene signaling pathway has been intensively studied for over four decades, and as a result of these efforts, a linear model of ethylene signal perception and conduction now dominates. The first component of this signaling machinery is the ethylene receptors, multidomain proteins similar to bacterial receptor histidine kinases, localized in the endoplasmic reticulum. Knowledge of the fine structural organization of receptors is certainly an important step in understanding their receptor function.

Obtaining protein crystals is one of the powerful tools for studying the fine functional organization of proteins, including receptor proteins. The difficulties associated with the crystallization of membrane proteins, such as the ethylene receptors ETR1, ETR2, ERS1, ERS2, EIN4, are well known.

The authors used the well-established method of crystallization in lipid mesophases. The authors managed to obtain crystals of recombinant proteins ETR1 (aa 1-316) and ETR2 (aa 1-186), containing transmembrane domains. However, of the many crystals obtained in their extensive screening, only two conditions gave crystals of ETR1 (aa 1-316) and ETR2 (aa 1-186 mT2) with clear but blurred diffraction.

It may be worth trying crystallization in the presence of ethylene or an inhibitor of its binding to the receptor, 1-MCP.

In general, despite the far from final result, this work deserves publication. There is a way to solve the problem.

In this world, not everything is so simple.

Author Response

We thank the reviewer for these encouraging comments on our work and for the suggestion to include ethylene or antagonist 1-MCP in our LCP crystallization.

Reviewer 3 Report

Comments and Suggestions for Authors

Re: biomolecules-2888780

Rüffer et al presented a manuscript entitled “Crystallization of Ethylene Plant Hormone Receptor Screening for the Structure” for publication consideration with Biomolecules. 

The paper highlighted plant hormone ethylene is a key regulator of plant growth, development and stress adaptation, it pointed out its importance to global agriculture. Ethylene perception and response are mediated by a family of integral membrane receptors (ETRs) which form dimers and higher-order oligomers in their functional state determined by the binding of Cu(I) as a cofactor to their transmembrane helices at the ER-Golgi endomembrane system. Author reported here a new methodology which use crystallization of transmembrane (TM) and membrane-adjacent domains of plant ethylene receptors by the Lipidic Cubic Phase (LCP) technology using vapor diffusion in meso crystallization. 

The manuscript provided new technique of diffraction-quality crystals for study role of 

protein stabilization in plant. It is scientifically sound research and interesting. Authors presented a well-designed study. 

There are some minor issues in presentation such as in “Figure 5. Crystal pictures under UV light”, the scale bars depict 100 µm length but they are all in different size in the figure, showed in 6 scales. Please try to address these. 

Author Response

Thank you for your positive feedback and for bringing to our attention the size variations of the scale bars in Figure 5. We have now added the microscope magnifications used to the figure to address this issue.